**Data Availability Statement:** All data generated in this study may be found on our github: https://

# UV decontamination of personal protective equipment with idle laboratory biosafety cabinets during the COVID-19 pandemic

Davis T. Weaver[1]☯, Benjamin D. McElvany[2]☯, Vishhvaan Gopalakrishnan[1]☯, Kyle J. Card[1,8], Dena Crozier[1], Andrew Dhawan[1,3], Mina N. Dinh[1], Emily Dolson[1], Nathan Farrokhian[1], Masahiro Hitomi[1], Emily Ho[1], Tanush Jagdish[4], Eshan S. King[1], Jennifer L. Cadnum[9], Curtis J. Donskey[9], Nikhil Krishnan[1], Gleb Kuzmin[1], Ju Li[7], Jeff Maltas[5], Jinhan Mo[6], Julia Pelesko[1], Jessica A. Scarborough[1], Geoff Sedor[1], Enze Tian[7], Gary C. An[2], Sean A. Diehl[2]*, Jacob G. Scott[1]*

**1** Cleveland Clinic Lerner Research Institute and Case Western Reserve University School of Medicine, Cleveland, OH, United States of America, **2** University of Vermont Medical Center, Burlington, VT, United States of America, **3** Cleveland Clinic, Division of Neurology, Cleveland, OH, United States of America, **4** Dana Farber Cancer Insitute, Harvard University, Boston, MA, United States of America, **5** University of Michigan, Ann Arbor, MI, United States of America, **6** Tsinghua University, Beijing, China, **7** Massachusetts Institute of Technology, Cambridge, MA, United States of America, **8** Michigan State University, East Lansing, MI, United States of America, **9** Louis Stokes Cleveland VA, Cleveland, United States of America

☯ These authors contributed equally to this work.
* Sean.Diehl@med.uvm.edu (SAD); scottj10@ccf.org (JGS)

## Abstract

Personal protective equipment (PPE) is crucially important to the safety of both patients and medical personnel, particularly in the event of an infectious pandemic. As the incidence of Coronavirus Disease 2019 (COVID-19) increases exponentially in the United States and many parts of the world, healthcare provider demand for these necessities is currently outpacing supply. In the midst of the current pandemic, there has been a concerted effort to identify viable ways to conserve PPE, including decontamination after use. In this study, we outline a procedure by which PPE may be decontaminated using ultraviolet (UV) radiation in biosafety cabinets (BSCs), a common element of many academic, public health, and hospital laboratories. According to the literature, effective decontamination of N95 respirator masks or surgical masks requires UV-C doses of greater than 1 $Jcm^{-2}$, which was achieved after 4.3 hours per side when placing the N95 at the bottom of the BSCs tested in this study. We then demonstrated complete inactivation of the human coronavirus NL63 on N95 mask material after 15 minutes of UV-C exposure at 61 cm (232 $\mu Wcm^{-2}$). Our results provide support to healthcare organizations looking for methods to extend their reserves of PPE.

## Introduction

Personal protective equipment (PPE) is essential for protecting medical personnel and patients during outbreaks of airborne or droplet borne infectious diseases. In particular, the use of face shields, surgical masks and N95 respirators are recommended for infections that may be

github.com/TheoryDivision/covid19_biosafety_cabinet.

**Funding:** The author(s) received no specific funding for this work.

**Competing interests:** JGS, the senior author, has, subsequent to this research, led a related patent for a UV decontamination device (PCT/US2021/022771, led march 17, 2021: \Decontam-ination System"). This does not alter our adherence to all PLOS ONE policies on sharing data and materials, and in fact, the patent was a reaction to this work, and so therefore was entirely performed subsequently to this research. Further, beyond the fact that the patent is for a UV-C decontamination chamber, it has little to do with this research.

transmitted by respiratory droplets or airborne particles [1]. Due to the rapidly emergent nature of the novel Coronavirus Disease 2019 (COVID-19) and stringent requirements of proper PPE protocol, many hospitals are running dangerously low on these protective devices. As a result, both patients and their healthcare providers are at increased risk of contracting and spreading SARS-CoV-2, the virus that causes COVID-19.

As previously suggested, one method of preserving our current supply of PPE is through cycles of decontamination and reuse with ultraviolet germicidal irradiation (UVGI). Substantial work has been done evaluating the efficacy of UVGI for decontamination of N95 filtering faceplate respirators (FFRs) [2–6]. Recently, UVGI has also been used to facilitate decontamination and re-use of plastic face shields [7]. Ultraviolet (UV) light is a form of electromagnetic radiation which contains more energy than visible light, but less energy than x-rays. It can be categorized into UV-A (315–400 nm), UV-B (280–315 nm), and UV-C (100–280 nm). The germicidal effectiveness of UV radiation is in the 180–320 nm range, with a peak at 265 nm [8]. The higher-energy UV-C rays can damage DNA and RNA via cross-linking of thymidine and uracil nucleotides, respectively, thus preventing the replication of microbes such as bacteria and viruses [9]. At these wavelengths, the amount of surface pathogen inactivation is directly proportional to the dose of UV radiation, with dosage being defined as the product of intensity (W/m$^2$) and exposure duration(s) [10, 11]. Therefore, UVGI is a relatively simple method of decontamination that causes minimal damage to the respirator and avoids the use of irritating chemicals.

One potential concern with using UVGI decontamination of N95 masks is the possibility of material degradation and reduced filtration efficacy. Multiple studies have addressed this question and overall found no significant deleterious effect of UV irradiation on the integrity and filtration capacity of several medical-grade masks [4, 5, 12, 13]. Their results are summarized in Table 1.

There are two primary types of damage that can happen to an N95 mask: 1) structural damage that affects fit, and 2) damage to the filter. Structural damage can be readily detected by performing regular respirator fit tests. Thus, assuming fit tests are performed regularly, the possibility of damage to the filter is the greater concern because it cannot be detected as easily.

**Table 1. Key findings from research on UV-mediated mask degradation.**

| Study | Total dose of UV radiation used | Results | Masks tested |
|---|---|---|---|
| Lore et al., 2012 | 1.8 Jcm$^{-2}$ | "No significant degradation in filter performance at 300-nm particle size." | 3M 1860s and 3M 1870 |
| Lindsley et al., 2015 | 120 Jcm$^{-2}$—950 Jcm$^{-2}$ | Essentially no effect on flow resistance. Some mask types showed increased particle penetration at higher doses. Bursting strength of some filter layers decreased with higher doses. Strap breaking strength decreased substantially at high doses. At 120 Jcm$^{-2}$ the only significant degradation was decreased bursting strength on one filter layer of one mask. | 3M 1860, 3M 9210, Gerson 1730, and Kimberly-Clark 46747 |
| Viscusi et al., 2009 | 3.24 Jcm$^{-2}$ (half to each side of the mask) | No effect on filter penetration, airflow resistance, or physical appearance. | Three N95 FFR models, three surgical N95 respirator models, and three P100 models. The N95s were randomly selected from the US Strategic National Stockpile and the P100s were randomly selected from commercially available models. |
| Bergmann et al., 2010 | 4.68 Jcm$^{-2}$ | "[No] observable physical changes" | Same as Viscusi et al., 2009 |
| Heimbuch, 2019 | 1 Jcm$^{-2}$ to 20 Jcm$^{-2}$ applied in cycles of 1 Jcm$^{-2}$ | Fit test performance not significantly affected by UVGI but is affected by repeated doffing and donning. Minor effect on filtration efficiency for one mask after 10 Jcm$^{-2}$ of UV radiation, but still within safe limits. Overall, no "meaningful" effect. | 3M 1860, 3M 1870, 3M VFlex 1805, Alpha Protech 695, Gerson 1730, Kimberly-Clark PFR, Moldex 1512, Moldex 1712, Moldex EZ-22, Precept 65–3395, Prestige Ameritech RP88020, Sperian HC-NB095, Sperian HC-NB295, U.S. Safety AD2N95A, and U.S. Safety ADN95 |

The only study to observe either type of damage used a range of very high doses of UVGI [4]. At their lowest dose (120 Jcm$^{-2}$), the only significant damage was that, for one model of mask, one layer of the filter became significantly more susceptible to being punctured by a steel ball (decreased burst strength). At higher doses damage gradually became more significant.

Based on these studies, UV radiation appears to be safe for N95 masks at the levels necessary to achieve decontamination. The decision-making challenge is to determine a safe upper limit on the number of decontamination cycles an individual mask experiences, as damage from UV radiation is cumulative. 4.68 Jcm$^{-2}$ is the highest total amount of UV radiation for which absolutely no physical degradation was observed. In a desperate situation (e.g. where the alternative is not decontaminating or using no PPE), up to 20 Jcm$^{-2}$ or perhaps even 120 Jcm$^{-2}$ may be safe. Note that repeated donning and doffing of masks also leads to structural damage [14]. It is likely that masks would need to be replaced for this reason well before they experienced enough decontamination cycles to experience a cumulative UV dose of 20 Jcm$^{-2}$.

Although there is no current consensus on the amount of UV radiation required to inactivate SARS-CoV-2, the UV dose required to inactivate 90% of single-stranded RNA viruses on gel media is an estimated 1.32—3.20 mJcm$^{-2}$ [2]. These estimates represent the likely dose needed to inactivate SARS-CoV-2 on face shields, while porous materials like N95 masks or surgical masks present a different challenge. Several studies have been conducted to identify the required dose to inactivate other single-stranded RNA viral contaminants on N95 masks. For example, for a 3 log reduction in recovered MS2 phage particles placed on soiled FFR masks, Vo et al. found a necessary UVGI dose of 4.32 Jcm$^{-2}$ [15]. Comparably, for a variety of mask models, Mills et al. found that a 1 Jcm$^{-2}$ UVGI dose conferred a range of 1.42 to 4.84 log reduction of H1N1 influenza viral load [3]. While more *in vitro* studies are likely needed to identify the dose required for safe decontamination, literature suggests that a dose of at least 1Jcm$^{-2}$ is required to decontaminate soiled FFR masks prior to re-use. These data are summarized in a recently released CDC report [16]. UVGI and other decontamination methods are also summarized online at https://www.n95decon.org.

Many university-affiliated hospitals and higher academic laboratories have access to biosafety cabinets (BSCs) that are regularly used in research to decontaminate laboratory equipment via UV-C light. Due to current social distancing and quarantine measures, there likely exist a substantial number of BSCs that are not currently in use and therefore may be available to be temporarily repurposed for N95 respirator, or other PPE decontamination. While this paper focuses on BSCs, many other promising approaches to UVGI decontamination are being designed by other groups [7, 17].

Given the urgency of the ongoing COVID-19 pandemic, we sought to determine if BSCs could be temporarily repurposed for UVGI decontamination to preserve a dwindling supply of PPE. To do this, we measured the minimum light intensity output by a standard BSC, as well as the variability of light intensity between and within several BSCs. From these measurements, we calculate a recommended time of 4.3 hours per side (62 minutes per side if the masks can be elevated to 19 cm from the UV-C source) to irradiate FFRs in a BSC to inactivate potential SARS-CoV-2 virus, or 20 minutes per side to irradiate solid PPE, like face shields.

## Methods

Three different class II type A2 BSCs were used in this experiment, the LabGard ES NU-540–400 Class II, Type A2 model (NuAire, Plymouth, MN), the Labgard ES ENergy Saver Class II, Type A2 model (NuAire, Plymouth, MN), and the ThermoFisher Model 1377 Type A2. The LabGard BSCs were equipped with a General Electric Germicidal Lamp model G30T8, which is reported to use 253.7 nm UV-C radiation and provide an average intensity of 100 $\mu W$cm$^{-2}$

to the cabinet floor. The ThermoFisher BSC was equipped with an Atlanta Ultraviolet 254 nm bulb.

## UV meter measurements

We measured UV fluence using a UV meter (to obtain absolute measurements) and measured variance due to mask geometry using an array of three photodiodes. Experiments were performed on N95 3M 1860S respirators. These measurements were conducted by placing a UV fluence meter (General Tools UV512C) at each of nine positions in each BSC (see **S2 Fig** in S1 File). Measurements were also taken in each of the 9 positions at elevations of 33 cm and 48.3 cm above the BSC floor. The UV meter was left in place until the reported value stabilized, at which point that value was recorded as the quantity of UV radiation reaching that position in the BSC. An array of measurements were also taken using photodiodes to assay heterogeneity within a given position, as well as in several other BSCs (see **S3 Fig** in S1 File).

## Experimental UV-C inactivation of human coronavirus NL63-contaminated N95 mask material

We performed experiments to assess the ability of laboratory BSCs to inactivate the NL63 human coronavirus. A Staples brand hole punch was used to make 0.5 cm punches from a 1860 3M™ N95 Mask. These punches were placed into wells of 3524 Corning Costar 24-welled plates with the exterior surface of the mask (blue) facing up. 25 uL of $4.6x10^5$ TCID50/mL NL63 human coronavirus was pipetted onto the blue surface of the punches. The punches were then either not exposed to or exposed to UV from a biosafety cabinet UV bulb (Atlantic Ultraviolet 05–0660) for different amounts of time. UV exposure occured 24" directly below the UV bulb at a dose of 232 uW $cm^{-2}$. At the appropriate time, the punches were washed with the same 1 mL of virus infection media five times into its well by pipetting the 1 mL with a P1000 pipettor directly into the middle of the punch. The virus infection media was then immediately used to determine its titer by TCID50 on LLC-MK2 (ATCC CCL-7.1) cells. The TCID50 was calculated using the Reed and Muench method [18]. Virus Titrations were performed by end-point titration in LLC-MK2 cells. Cells were inoculated with 100uL in 10-fold serial dilutions of the virus infection media taken from the mask punch wells and incubated at 34 C plus 5% CO2. After one hour, an additional 500uL of virus infection media added to wells. Plates incubated at 34 C plus 5% CO2 and cytopathic effect were scored until the same score was seen three days in a row (Day 12).

## Results

### UV-C measurements in multiple BSCs

To evaluate the feasibility of using a BSC for UV-C irradiation-based decontamination of PPE we measured absolute UV-C radiation at different areas across the working surface of three different BSC units. Our measurements show a clear pattern of spatial variation in UV intensity (see Fig 1). Interestingly, many of the measured values substantially exceed the manufacturer's specified fluence (100 $\mu W$cm$^{-2}$). In BSC 1, all of the measurements were greater than 100 $\mu W$cm$^{-2}$. Because the UV meter cannot be attached to a mask, these measurements do not take into account variation produced by mask geometry.

Importantly, the minimum observed value differed substantially between BSCs: 111 $\mu W$cm$^{-2}$ vs. 64 $\mu W$cm$^{-2}$. This finding is consistent with the fact that the amount of UV-C light emitted is known to decay as bulbs age, and highlights the importance of either using new bulbs or measuring UV-C output to verify that it is sufficient. Note that annual BSC

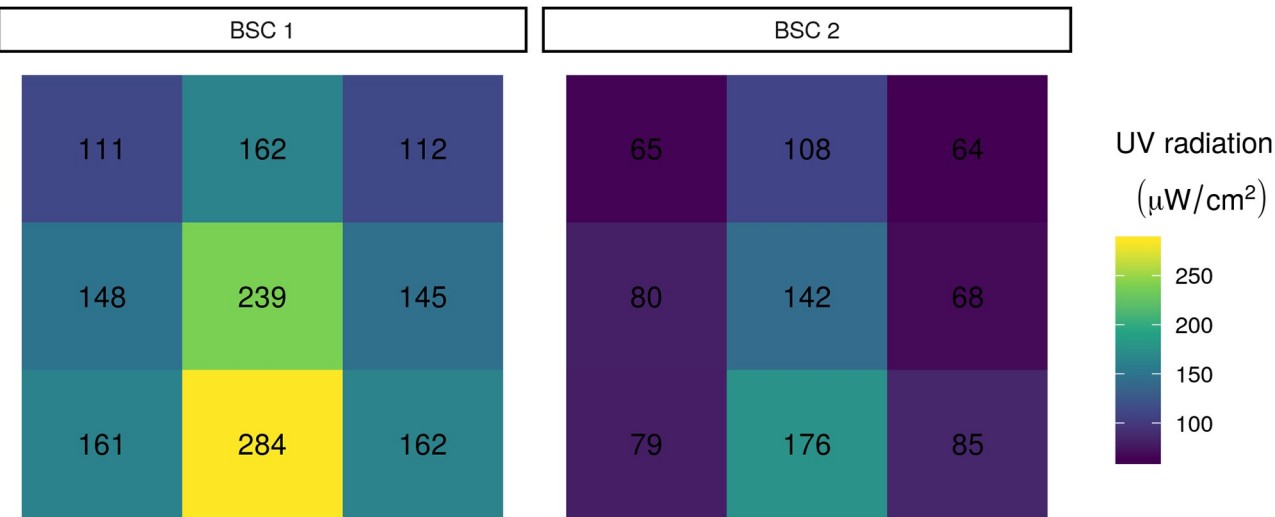

**Fig 1. UV radiation in each sector of each BSC as measured with a UV meter.** Each of the nine sections per BSC shows the UV radiation measured in the section. Numbers indicate UV radiation measured in each section.

certification (NSF Standard 49) does not include measuring UV output, although many certification agencies offer it as an optional add-on test.

**Elevated measurements.** Given a cylindrical UV source with length roughly on the same order of magnitude as the distances from which in intensity is measured, we expect that time for desired dose will increase at least faster than linearly with respect to distance from UV lamp [19]. To assess the possibility of raising masks within the BSC to reduce decontamination time based this relationship, we also took measurements of UV intensity at 33 cm and 48.3 cm above the BSC floor (Fig 2). The total height of the BSC was 67.3 cm.

Indeed, our UV intensity data with respect to the nearest distance to the UV bulb, stratified by position relative to the length of the UV lamp, reveals a close fit to an inverse square function function. These data suggest that raising the object to be decontaminated towards the UV-C source allows for delivery of much higher doses than those achieved on the floor of the BSC (Fig 3).

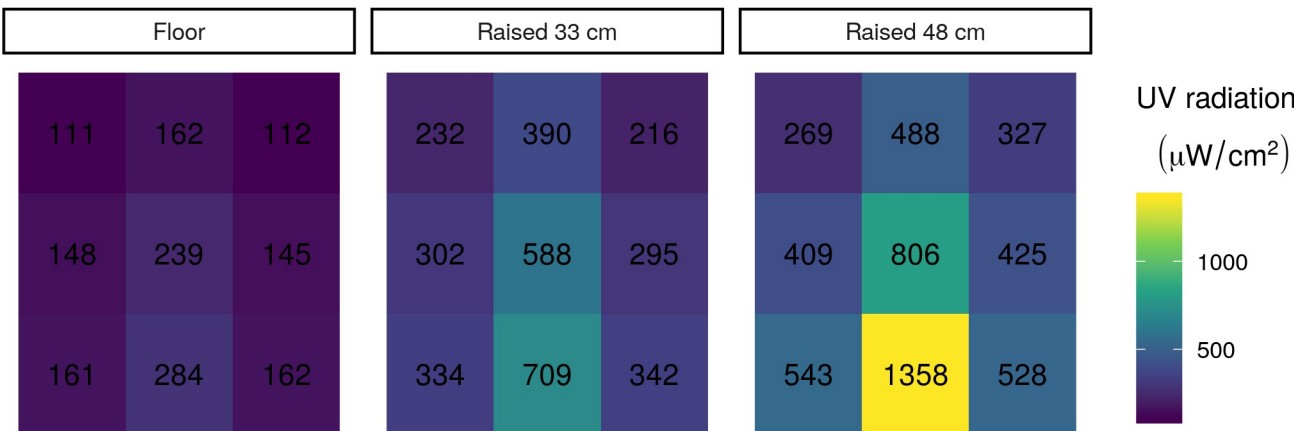

**Fig 2. UV radiation in each sector of BSC 1 at three different heights.** Each of the nine sections per elevation shows the UV radiation measured in the section. Numbers indicate UV radiation measured in each location with the UV meter.

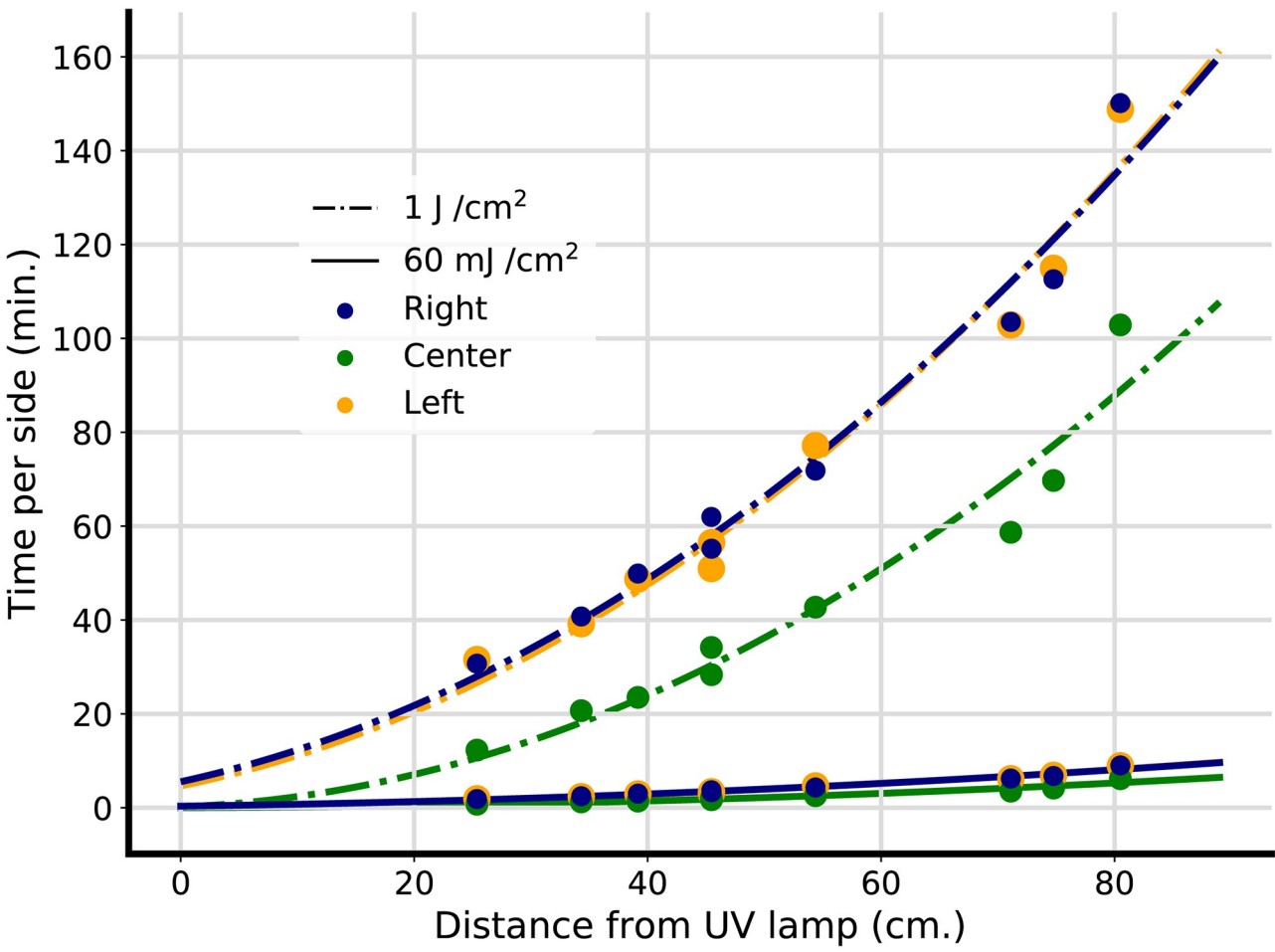

**Fig 3. Time to decontaminating dose with respect to distance from UV lamp for face-shield and FFR decontaminating doses.** An inverse square function was fit to UV fluence data from hood 1 at various heights for the left, center, and right-hand sections of the BSC, as visualized in Fig 2, and used to calculate time for decontaminating dose per side at target doses of 1 $cm^{-2}$ and 60 mJ $cm^{-2}$ This approximate inverse square relation can be exploited to deliver high doses of UV within a BSC in a reasonable amount of time by positioning PPE close to the UV lamp.

The literature on this subject, including a recent CDC summary, suggests that a dose of *at least* 1 Jcm$^{-2}$ of UV-C is required to decontaminate FFRs [3, 14–16]. Hospitals can, of course, choose a different target dose based on their internal risk analysis. To estimate the time (per side of the mask) required for decontamination in a BSC, we can use the following equation:

$$\frac{\text{target dose mJ}}{\text{cm}^2} \times \frac{\text{cm}^2}{\text{min. intensity}\mu W} \times \frac{1000 \mu W \text{ seconds}}{1 \text{ mJ}} \times \frac{1 \text{ minutes}}{60 \text{ seconds}} = \text{ recommended time. (1)}$$

For explanations of all terms in this equation, see Table 2. Selecting 1 J cm$^{-2}$ as our target dose,

**Table 2. Description of equation terms.**

| Value | Description |
|---|---|
| target dose | UV dose required to achieve desired level of decontamination (using 1 J cm$^{-2}$) |
| min. intensity | The lowest UV-C intensity anywhere in the BSC in $\mu W$cm$^{-2}$ |
| recommended time | Estimated time (in minutes) to decontaminate one side of an FFR |

this equation reduces to:

$$\frac{1000 \text{ minutes}}{\text{min. intensity}} = \text{ recommended time (minutes).} \qquad (2)$$

Now we must choose a value for intensity. To ensure that all masks in the BSC achieve the target UV radiation dose, we must select the minimum level of UV-C radiation anywhere in the BSC. Based on the UV meter data, the lowest UV-C radiation level we observed across both hoods is 64 $\mu W\text{cm}^{-2}$. Plugging these values into Eq 2, we find that the minimum time required to decontaminate FFRs in a standard BSC, assuming the variance we measured above, is 4.3 hours per side. As that may be a prohibitively long time to wait, we also consider the possibility of elevating PPE within a BSC to reduce the decontamination time. Based on our measurements in Fig 2, we estimate that raising PPE 48.1 cm off the floor of a 67.3 cm tall BSC with a specified fluence of $100\mu W$ should reduce the needed decontamination time to a minimum of 62 minutes per side, given the lowest UV measurement made at that height.

### Estimating time to decontaminate face-shields in a BSC

In order to decontaminate face-shields in a BSC, much lower UV doses are sufficient. 2–5 mJ $\text{cm}^{-2}$ of UV radiation is estimated to kill most single-stranded RNA viruses on gel media (similar to the hard plastic face-shield). To err on the side of caution and ensure that other pathogens were also deactivated, we will base our recommendation for face-shield decontamination on a target dose of 60 mJ $\text{cm}^{-2}$. Because of the flat, uniform nature of face-shields, we also do not need to account for UV dose variation due to mask geometry. As a result, we can use the following equation to calculate our recommended decontamination time:

$$\frac{\text{target dose mJ}}{\text{cm}^2} \times \frac{\text{cm}^2}{\text{min intensity}\mu W} \times \frac{1000\mu W \text{ seconds}}{1 \text{ mJ}} \times \frac{1 \text{ minutes}}{60 \text{ seconds}} = \text{ recommended time (minutes).} \quad (3)$$

Plugging in 60 mJ $\text{cm}^{-2}$ as our target dose, and 64 $\mu W\text{cm}^{-2}$ as the minimum intensity, we calculate a recommended time in the bottom of our BSC of 15.6 minutes per side for plastic face-shield decontamination.

### Virologic validation

We found that 5 minutes of UV-C radiation (232 W·cm–2) reduced recovery of infectious NL63 virus from the exterior of N95-mask material by over 3 logs and complete inactivation was achieved after 15 minutes (Fig 4).

### Discussion

Ideally, a new mask or respirator would be used for each individual to minimize the transmission of infectious diseases that are airborne or transmitted via respiratory droplets. However, crises such as the current COVID-19 pandemic can create shortages that necessitate measures to conserve PPE. Among potential methods for decontamination, previous work has suggested UVGI results in less physical deformation than bleach, microwave irradiatin, and vaporized hydrogen peroxide [5].

Additionally, this and other investigations of UVGI for the purpose of PPE decontamination was motivated by the ubiquity of UV lamp equipped biosafety cabinets, especially at large biomedical research institutions. Various groups have therefore begun decontaminating respiratory protective equipment themselves using UVGI and "homebrew" setups. For example,

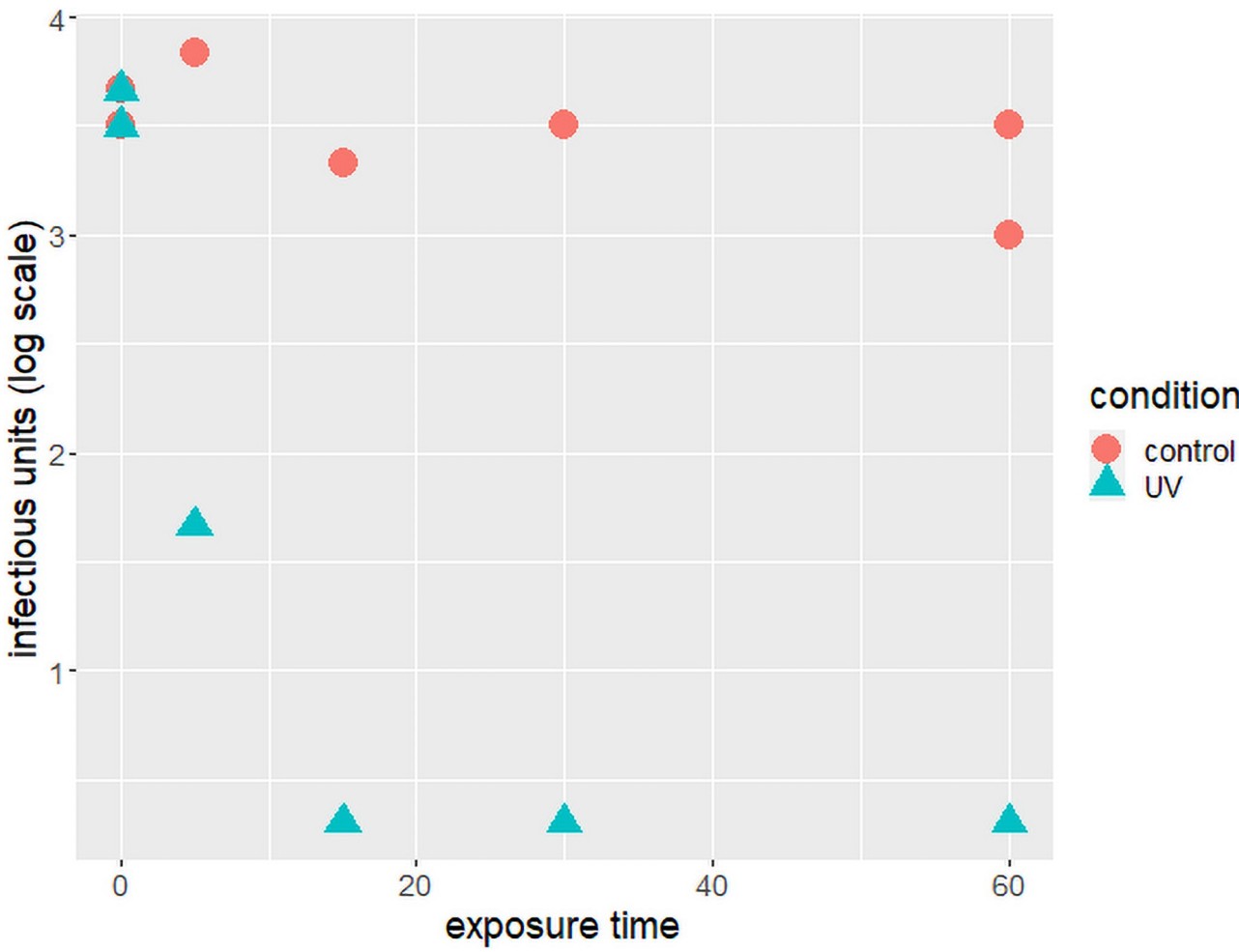

**Fig 4. Infectious units recovered in UV exposed versus control masks strips.** 20 minutes of UV-C radiation in a BSC was sufficient to achieve more than a 3 log reduction in viral recovery of the NL63 coronavirus.

enterprising clinicians at the University of Nebraska Medical Center are stringing N95 respirators between two towers of UVGI bulbs placed on either side of a room in order to inactivate potential SARS-CoV-2 viral contaminants on the masks [17].

From our measurements, normalized to the technical specifications of the manufacturer using a typical BSC, we estimate the minimum time to decontaminate FFR is 4.3 hours per side. We estimate the minimum time to decontaminate face-shields is 15.6 minutes per side We invite other scientists to add measurements from their own BSCs to our github repository to allow continued updating of this recommendation [20]. Ideally, clinical sites interested in using this protocol should take measurements using calibrated UV fluence detectors of their specific BSCs prior to implementation of this protocol. If a calibrated UV detector is unavailable, UV test strips could provide an affordable way to ensure an appropriate UV dose is achieved in a given BSC. To calculate a time for an arbitrary BSC model, we recommend using Eq 2. In the future, it may be possible to design a technique that avoids the need to flip masks over and irradiate each side separately. By elevating masks off the surface of the BSC and, if necessary, placing reflective material underneath them, it should be possible to ensure that UV

radiation reaches the entire mask surface simultaneously and would reduce the manual labor and time required for this protocol.

Inspired by the protocol developed by Lowe et al., we propose a workflow to optimize the utilization of institutional resources [17].

1. Prior to use, PPE should be directly labeled to identify the original owner by both name and department.

2. After use, place in sealed packaging and distribute to BSC locations.

3. Using sterile technique, remove PPE from packaging and place on working surface of cabinet.

4. Ensure that there is no overlap of adjacent masks (including straps/headbands), as any unexposed areas will not be decontaminated.

5. After transfer, adequately decontaminate any external surface that came in contact with the used masks or packaging and destroy the packaging via biological waste.

6. For FFR: Close the hood and power on the UV light for 62 minutes on an elevated platform or 4.3 hours if the FFR is placed on the floor of the BSC.

7. For face-shields: Close the hood and power on the UV light for 15.6 minutes

8. After this duration, power off the UV light, open the cabinet, and carefully flip the masks to expose the opposite side, ensuring no overlap of adjacent masks.

9. Close the hood and power on the UV light again for the recommended time for your PPE type.

10. Again, adequately decontaminate or dispose of any external surface that comes in contact with the masks.

11. Once the full duration has elapsed, power off the UV light and open the hood.

12. While maintaining sterility of the cabinet, add a tally to each mask indicating the number of UVGI cycles it has experienced and individually place in sterile, sealed packaging.

13. Remove packages from cabinet and redistribute to original owner.

## Limitations

Despite the measures taken here to ensure adequate decontamination of PPE, following this protocol by no means guarantees complete sterilization or decontamination. This method should be implemented *only if* PPE *must be reused*. FFRs contain multiple layers of filtration, and respiratory droplets may penetrate into the inner layers. Though UV-C light has been shown to transmit into and through FFR materials, the transmittance of light ranges from 23–50% through the outer layer depending on the model of the FFR [6]. Therefore, the ability for UVGI to thoroughly sanitize FFRs may vary based on the ability for UV-C light to penetrate through to the internal filtering medium, which contributes the most filtration ability. Virologic testing to determine the degree of decontamination of the inner mask layers is ongoing.

Previous *in vitro* studies imply that the shape of the inactivation-curve is modulated by the surface being decontaminated. Generally, studies find a much lower dose needed to inactivate virus on gel or plate-based media compared to FFRs such as the N95 mask [2, 3]. The feasibility of our approach for decontaminating FFRs is therefore limited by the long-time duration (at least 4.3 hours per side) needed to achieve a germicidal UV-C dose on the floor of a BSC.

Variance in received dose due to the shape of the FFRs may also contribute to incomplete decontamination. We considered this possibility using an array of photo-diodes affixed to different positions on each mask throughout our 3 X 3 grid. In the areas of the grid receiving the lowest intensity (the front corners), the median observed proportional variance (max intensity/min intensity) across the masks was 2.17. Scaling our recommendation by this value, 9.4 hours per side would be required to decontaminate each mask. We did not incorporate this into our main recommendation due to concerns about the our use of directional sensors to measure received dose (i.e. the measured intensity varied substantially with direction of the sensor in addition to sensor position). We believe that our measurements with a UV fluence meter are more reliable and repeatable. We present the photo-diode measurements here as an important potential limitation and something that hospital systems should consider when calibrating their own BSCs. The full photo-diode data and results can be found in the S1 File.

Additionally, without measuring the absolute UV-C levels in a given BSC, it is not possible to be sure that it is outputting the specified amount of radiation. For instance UV-C lamps can produce visible light without a significant loss of intensity while UV intensity has fallen below the germicidal threshold. Ideally, UV-C fluence in each BSC should be measured and verified before using this protocol. Given the scarcity of UV-C fluence meters, however, this may not be possible in all cases. The next best solution is to use the newest UV-C bulbs available. Bulbs should be inspected and cleaned regularly to ensure that debris is not blocking UV radiation [21, 22]. With only three BSCs measured, we cannot fully quantify the amount of variation we expect to see across the set of all BSCs. There almost certainly exist BSCs with locations where the UV radiation received is lower than the lowest value we measured.

As discussed in the background, UV-C-mediated degradation of polymers within the respirator is another possible concern. Fit and filtration testing of the N95 respirators used in this experiment did not reveal any decline in filtration efficiency following UV-C exposure (**S4 Fig** in S1 File). While we do not anticipate such degradation being the limiting factor, we recommend that hospitals employing this approach take additional precautions such as: 1) labeling N95 respirators so that they can be reused by the same individual, 2) marking the number of times the same mask has undergone decontamination, as was recommended by Lowe et al. [17], and ensuring this number does not exceed 40, and 3) regularly fit-testing respirators. While our initial virology experiments were extremely promising, it should be noted that the conditions that were tested likely do not correspond to a heavy viral exposure to the masks, particularly the interior of the mask. Further, now that the pandemic has somewhat subsided in the US (post-peer review), and laboratories have reopened, we sought to recapiulate our initial results. Please see the S1 File for three additional experiments done which support the aforementioned claims for most viruses. As such, these results suggest that BSCs can safely decontaminate masks following low-titer viral exposure, as might occur during a routine encounter with an infectious patient. These results do not say anything about the ability of BSCs to decontaminate soiled masks.

## Public health implications

We believe that the presented method for decontamination of PPE using UVGI available through idle BSCs is a versatile and scalable method suitable for individual, or widespread institutional implementation. We estimate that there is an adequate abundance of idle BSCs at biomedical research institutions across the nation, most of which are idle given research hiatus due the current pandemic, which would allow for widespread use of BSCs for PPE decontamination The WHO estimated that roughly 89 million FFRs, 76 million, 1.6 million gloves will be needed internationally per month in response to the COVID-19 per month [23]. In order to

meet these demands, the worldwide production would need to increase by 40%. However, we suggest that a significant proportion of the international need for PPE can be met through the use of idle BSCs for decontamination.

## Supporting information

**S1 File.**
(PDF)

## Acknowledgments

Thanks to Tyler Cassidy, Jessica Cunningham and Lydia Kisley for their help. Additionally, we thank Amy Herr, Gary An, and Andrea Armani for their helpful conversations and comments. We would also like to thank everyone who supported this work with their encouraging tweets. In particular, we thank Mohamed Abazeed for his helpful comments on Twitter.

## Code and Data Availability

All data used in this paper and code written to analyze it are open source and publicly available [20].

## Author Contributions

**Conceptualization:** Davis T. Weaver, Vishhvaan Gopalakrishnan, Kyle J. Card, Dena Crozier, Andrew Dhawan, Mina N. Dinh, Emily Dolson, Nathan Farrokhian, Masahiro Hitomi, Emily Ho, Tanush Jagdish, Eshan S. King, Jennifer L. Cadnum, Curtis J. Donskey, Nikhil Krishnan, Gleb Kuzmin, Ju Li, Jeff Maltas, Jinhan Mo, Julia Pelesko, Jessica A. Scarborough, Geoff Sedor, Enze Tian, Gary C. An, Sean A. Diehl, Jacob G. Scott.

**Data curation:** Davis T. Weaver, Benjamin D. McElvany, Vishhvaan Gopalakrishnan, Mina N. Dinh, Tanush Jagdish, Jennifer L. Cadnum, Curtis J. Donskey, Nikhil Krishnan, Jessica A. Scarborough, Enze Tian.

**Formal analysis:** Davis T. Weaver, Benjamin D. McElvany, Vishhvaan Gopalakrishnan, Emily Dolson, Eshan S. King, Jennifer L. Cadnum, Curtis J. Donskey, Nikhil Krishnan, Enze Tian.

**Investigation:** Davis T. Weaver, Benjamin D. McElvany, Vishhvaan Gopalakrishnan, Emily Dolson, Masahiro Hitomi, Tanush Jagdish, Eshan S. King, Nikhil Krishnan, Jessica A. Scarborough, Jacob G. Scott.

**Methodology:** Davis T. Weaver, Benjamin D. McElvany, Vishhvaan Gopalakrishnan, Kyle J. Card, Dena Crozier, Andrew Dhawan, Mina N. Dinh, Emily Dolson, Nathan Farrokhian, Masahiro Hitomi, Emily Ho, Tanush Jagdish, Eshan S. King, Jennifer L. Cadnum, Curtis J. Donskey, Nikhil Krishnan, Gleb Kuzmin, Ju Li, Jeff Maltas, Jinhan Mo, Julia Pelesko, Jessica A. Scarborough, Geoff Sedor, Enze Tian, Jacob G. Scott.

**Resources:** Masahiro Hitomi, Jacob G. Scott.

**Software:** Davis T. Weaver, Vishhvaan Gopalakrishnan, Emily Dolson.

**Supervision:** Masahiro Hitomi, Ju Li, Jinhan Mo, Gary C. An, Sean A. Diehl, Jacob G. Scott.

**Validation:** Davis T. Weaver, Vishhvaan Gopalakrishnan.

**Visualization:** Davis T. Weaver, Vishhvaan Gopalakrishnan, Emily Dolson, Eshan S. King.

**Writing – original draft:** Davis T. Weaver, Emily Dolson, Eshan S. King, Nikhil Krishnan, Jessica A. Scarborough, Jacob G. Scott.

**Writing – review & editing:** Davis T. Weaver, Benjamin D. McElvany, Vishhvaan Gopalakrishnan, Kyle J. Card, Dena Crozier, Andrew Dhawan, Mina N. Dinh, Emily Dolson, Nathan Farrokhian, Masahiro Hitomi, Emily Ho, Tanush Jagdish, Eshan S. King, Jennifer L. Cadnum, Curtis J. Donskey, Nikhil Krishnan, Gleb Kuzmin, Ju Li, Jeff Maltas, Jinhan Mo, Julia Pelesko, Jessica A. Scarborough, Geoff Sedor, Enze Tian, Gary C. An, Sean A. Diehl, Jacob G. Scott.

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
