## [Decision Letter · Decision Letter 0]

1 Dec 2020

PONE-D-20-33184

UV Sterilization of Personal Protective Equipment with Idle Laboratory Biosafety Cabinets During the Covid-19 Pandemic

PLOS ONE

Dear Dr. Scott,

Thank you for submitting your manuscript to PLOS ONE. After careful consideration, we feel that it has merit but does not fully meet PLOS ONE’s publication criteria as it currently stands. Therefore, we invite you to submit a revised version of the manuscript that addresses the points raised during the review process. 

We look forward to receiving your revised manuscript.

Kind regards,

Albert J. Fornace Jr, MD

Academic Editor

PLOS ONE

Journal Requirements:

2.Thank you for stating the following in the Acknowledgments Section of your manuscript:

[While this specific project was not directly funded by any body, we would like to thank our

funders in the form of the National Institutes of Health and the American Cancer Society and the Taussig Cancer

and Lerner Research Institute.]

 [The author(s) received no specific funding for this work.]

3.We note that you have a patent relating to material pertinent to this article. Please provide an amended statement of Competing Interests to declare this patent (with details including name and number), along with any other relevant declarations relating to employment, consultancy, patents, products in development or modified products etc. Please confirm that this does not alter your adherence to all PLOS ONE policies on sharing data and materials, as detailed online in our guide for authors http://journals.plos.org/plosone/s/competing-interests by including the following statement: "This does not alter our adherence to  PLOS ONE policies on sharing data and materials.” If there are restrictions on sharing of data and/or materials, please state these. Please note that we cannot proceed with consideration of your article until this information has been declared.

4.We noticed you have some minor occurrence of overlapping text with the following previous publication, which needs to be addressed: https://www.ajicjournal.org/article/S0196-6553(20)30995-0/fulltext

The text that needs to be addressed involves the Introduction, the first couple paragraphs of the Discussion, and the Limitations section. In your revision ensure you cite all your sources (including your own works), and quote or rephrase any duplicated text outside the methods section. Further consideration is dependent on these concerns being addressed.

Additional Editor Comments (if provided):

Please refer to comments of Reviewer 2

Reviewers' comments:

Reviewer's Responses to Questions

**Comments to the Author**

1. Is the manuscript technically sound, and do the data support the conclusions?

Reviewer #1: Yes

Reviewer #2: Yes

2. Has the statistical analysis been performed appropriately and rigorously? 

Reviewer #1: I Don't Know

Reviewer #2: Yes

3. Have the authors made all data underlying the findings in their manuscript fully available?

Reviewer #1: Yes

Reviewer #2: Yes

4. Is the manuscript presented in an intelligible fashion and written in standard English?

Reviewer #1: Yes

Reviewer #2: Yes

5. Review Comments to the Author

Reviewer #1: This study validates the use of the UVC lamp that is standard in the back of biosafety cabinets for filtering facepiece respirator (FFR also referred to as N95 respirator) decontamination. The authors did a nice job quantifying the radiation dose within the space of the cabinet. They also provided guidance to the required exposure time to achieve acceptable decontamination. In addition, the authors validated UVGI as method to decontaminate face shields. The equations to calculate minimum exposure time were provided as well. The authors conclude that the biosafety cabinets that are common in microbiology labs are an effective method to decontaminate respirators. Overall, the study was well organized and followed the scientific method. The method and results were clearly described. However, additional work is needed in order to provide value to the reader. I will address the limitations of this study next.

Novelty: using biosafety cabinet UV light to decontaminate FFR was established by NIOSH/NPPTL lab (Viscousi et al., 2007 and 2009) in multiple publications, see for example reference 5 of this paper. This study provides more details on the method already established by Viscousi et al.

Method: cutouts of N95 respirators are not a good representation of a respirator. Decontamination for the purpose of reuse should be validated for the entire respirator which is three dimensional and has additional parts such as headbands and nose clip. Shadowing effect from headbands and respirator curvature were not studies. Headband placement during decontamination is key to ensure headbands are not covering parts of the respirator, thereby, preventing decontamination of those parts. The authors need to address headband placement during the process. One type of respirator was evaluated: 3M 1860 cup, this is one of several different geometries that include pleated and folded respirators. How does the shape of those models affect UVGI performance as disinfection method? I realize shortages of N95 respirators is a constraint, so the authors may not have access to other respirator geometries.

Typically, most of filtered pathogens (viruses and bacteria) are deposited in the filter layer of a respirator, not on the coverweb (top surface). N95 respirators have electret charged filter that captures pathogens, while the coverweb does not have that property, therefore, most pathogens (and particles) end up in the inner filter layers of a respirator. The authors of this study deposited droplets of virus inoculum on the top surface of the respirator cutout. This method does not simulate real use conditions, and validating virus deactivation on the surface layer does not correlate with full respirator decontamination. The authors indicate in the limitations section of the paper that virology results for the inner layers of the respirator are pending, I recommend waiting to include those results before publication as virus viability on the inner layers is significant since most filtered virus will be deposited in those inner layers.

Decontamination of face shield: it is not clear to me that UVGI is appropriate or practical to decontaminate face shields. I suggest the authors compare the efficacy of UVGI with soap and water rinse and disinfectant wipes. The key property that should be maintained after decontamination is the antifog performance of the face shield. Does UVGI maintain the antifog performance? How does it compare to washing with soap and water or using disinfectant wipes? Do those alternate methods adversely affect antifog performance of face shields?

Statistical analysis: at least three replicates per experiment should be conducted. It was not clear if replicates were done and how many, no error bars are shown on the graphs.

Need clarification: the literature recommendation of 1 J/cm2 exposure was cited. However, the authors did not clarify weather this is the total exposure (both sides of the respirator) or required exposure dose per side. Since a respirator is a three dimensional opaque device, to achieve effective disinfection it is necessary to expose outer and inner surfaces to the UV radiation.

Reviewer #2: This reviewer (also scientific editor for this manuscript) was favorable to publication of this report in PlosOne. Shortage of PPE's during the covid pandemic is a serious problem and the authors have shown that standard UV bulbs, which are used for sterilizing laminar flow hoods, are effective in inactivating the virus. Such equipment is commonly available. Single strand viruses are particularly sensitive to UV radiation but this needed to be confirmed in COVID-19 virus which the authors have accomplished. The authors have done a reasonable job on dosimetry and results are convincing. Potential problems with this approach are addressed such as aging of typical 254 nm UV bulbs. While they have discussed the challenge of the 3 dimensional nature of some PPE equipment and the issue of UV radiation penetration into filter material, these issues need to be elaborated on further.

6. PLOS authors have the option to publish the peer review history of their article (what does this mean?). If published, this will include your full peer review and any attached files.

Reviewer #1: **Yes: **Caroline Ylitalo

Reviewer #2: **Yes: **Albert J. Fornace Jr.

---

## [Author Response · Author response to Decision Letter 0]

26 May 2021

please see attached response to review

---

## [Decision Letter · Decision Letter 1]

21 Jun 2021

UV Sterilization of Personal Protective Equipment with Idle Laboratory Biosafety Cabinets During the Covid-19 Pandemic

PONE-D-20-33184R1

Dear Dr. Scott,

We’re pleased to inform you that your manuscript has been judged scientifically suitable for publication and will be formally accepted for publication once it meets all outstanding technical requirements.

Kind regards,

Albert J. Fornace Jr, MD

Academic Editor

PLOS ONE

Additional Editor Comments (optional):

Minor concerns have been addressed in the revision.

Reviewers' comments:

Reviewer's Responses to Questions

**Comments to the Author**

1. If the authors have adequately addressed your comments raised in a previous round of review and you feel that this manuscript is now acceptable for publication, you may indicate that here to bypass the “Comments to the Author” section, enter your conflict of interest statement in the “Confidential to Editor” section, and submit your "Accept" recommendation.

Reviewer #1: All comments have been addressed

Reviewer #2: All comments have been addressed

2. Is the manuscript technically sound, and do the data support the conclusions?

Reviewer #1: Yes

Reviewer #2: Yes

3. Has the statistical analysis been performed appropriately and rigorously? 

Reviewer #1: Yes

Reviewer #2: Yes

4. Have the authors made all data underlying the findings in their manuscript fully available?

Reviewer #1: Yes

Reviewer #2: Yes

5. Is the manuscript presented in an intelligible fashion and written in standard English?

Reviewer #1: Yes

Reviewer #2: Yes

6. Review Comments to the Author

Reviewer #1: The authors adequately addressed all my earlier comments and concerns. The additions and modifications made to the original manuscript increased the value and accuracy of this work. Well done!

Reviewer #2: The authors have addressed minor concerns in their review and paper is now acceptable for publication in my opinion.

7. PLOS authors have the option to publish the peer review history of their article (what does this mean?). If published, this will include your full peer review and any attached files.

Reviewer #1: **Yes: **Caroline Melkonian Ylitalo

Reviewer #2: No

---

## [Editor Report · Acceptance letter]

12 Jul 2021

PONE-D-20-33184R1 

UV Decontamination of Personal Protective Equipment with Idle Laboratory Biosafety Cabinets During the COVID-19 Pandemic 

Dear Dr. Scott:

I'm pleased to inform you that your manuscript has been deemed suitable for publication in PLOS ONE. Congratulations! Your manuscript is now with our production department. 

Kind regards, 

on behalf of

Dr. Albert J. Fornace Jr 

Academic Editor

PLOS ONE